# Balloon Dacryocystoplasty with Pushed Monocanalicular Intubation as a Primary Management for Primary Acquired Nasolacrimal Duct Obstruction

**DOI:** 10.3390/jpm13030564

**Published:** 2023-03-21

**Authors:** Chun-Chieh Lai, Cheng-Ju Yang, Chia-Chen Lin, Yi-Chun Chi

**Affiliations:** 1Department of Ophthalmology, National Cheng Kung University Hospital, College of Medicine, National Cheng Kung University, Tainan 704, Taiwan; 2Institute of Clinical Medicine, College of Medicine, National Cheng Kung University, Tainan 704, Taiwan; 3Department of Surgery, Chung Shan Medical University Hospital, Taichung 402, Taiwan; 4Department of Ophthalmology, Kaohsiung Medical University Hospital, Kaohsiung Medical University, Kaohsiung 807, Taiwan

**Keywords:** balloon dacryocystoplasty, monocanalicular intubation, Masterka tube, nasolacrimal duct obstruction, silicone tube intubation

## Abstract

Given the improvement in the instrument and techniques, novel surgical interventions emerged to avoid the osteotomy from the gold standard dacryocystorhinostomy (DCR) for treating primary acquired nasolacrimal duct obstruction (PANDO). This study’s aim is to compare the surgical outcomes of antegrade balloon dacryocystoplasty (DCP) with pushed monocanalicular intubation (MCI) to balloon DCP alone in patients with complete PANDO. Adult patients with complete PANDO receiving balloon DCP followed by pushed MCI or balloon DCP alone from December 2014 to May 2019 were retrospectively reviewed. A total of 37 eyes of 29 patients were treated with balloon DCP with pushed MCI for 1 month, whereas 35 eyes of 28 patients were treated with balloon DCP alone. The success rates at 1 month, 3 months, and 6 months after operation were 89.2%, 73.0%, and 70.2%, respectively, in balloon DCP with MCI group, and 62.9%, 62.9%, and 60.0%, respectively, in the balloon DCP alone group. The balloon DCP with pushed MCI group had a better success rate but only reached statistical significance at 1 month postoperatively (*p* < 0.01). Subgroup analysis was performed based on age. The success rate in those under 65 in the combined balloon DCP with MCI group was significantly higher than in balloon DCP alone group (72.7% vs. 9.1%, *p* = 0.004), whereas there was no significant difference between those aged at least 65 in the combined group and the balloon DCP alone group (69.2% vs. 83.3%, *p* = 0.2). Conclusively, there was no significant difference in the success rate between antegrade balloon DCP with and without pushed MCI in general. Nevertheless, the former procedure was associated with significantly higher surgical success rate than the latter in younger patients.

## 1. Introduction

Acquired nasolacrimal duct obstruction is the blockage of the lacrimal system, resulting in epiphora or excessive tearing, intermittent mucopurulent discharge, and mattering of the involved eye. It occasionally causes lacrimal sac mucocele, fistula, abscess, acute or chronic dacryocystitis, or even orbital cellulitis and cavernous sinus thrombosis [1]. It can be roughly classified as primary acquired nasolacrimal duct obstruction (PANDO) or secondary acquired lacrimal duct obstruction (SALDO) [1]. SALDO is secondary to various defined etiologies of lacrimal obstructions categorized into infectious, inflammatory, traumatic, mechanical, and neoplastic SALDO, and the treatments targeting the cause of the obstruction may efficiently relieve the symptoms, whereas PANDO is defined as nasolacrimal duct obstruction of unknown etiology.

Surgical interventions for PANDO can be simply divided into reconstructive surgery and bypass surgery [2]. Although dacryocystorhinostomy (DCR), which is categorized as bypass surgery, has been the gold standard treatment for PANDO due to its high success rate [3,4,5,6,7,8], there have been some attempts to avoid osteotomy and perform less invasive procedures to manage nasolacrimal duct obstruction, including balloon dacryocystoplasty (DCP) and silicone stent intubation, which are categorized as reconstructive surgeries. Current studies on balloon DCP, silicone stent intubation, or a combination of the two procedures are mostly focused on partial PANDO, where the reported success rates vary, ranging from 25% to 76% [9,10,11,12,13,14,15]. Some studies have discussed balloon DCP for complete PANDO [13,14,16], while limited studies have discussed silicone stent intubation [2,16,17], and even fewer studies reported the result of combination procedure in complete PANDO.

Nowadays, there are two main types of monocanalicular intubation (MCI) available for management of the nasolacrimal duct obstruction. One is a pulled-type MCI and the other is a pushed-type MCI. Compared with the pushed-type, the probe guide should be drawn out from the nostril through the inferior meatus in the pulled-type MCI, which potentially may lead to intra-operative bleeding due to the penetration of the mucosa of inferior meatus, and subsequent patients’ discomfort during and after the operation [18]. On the other hand, the probe guide is placed into a silicone tube in the pushed-type MCI, which may require no manipulation of the nostril and therefore decreases the risk of nasal bleeding during the procedure [19]. We therefore employed the pushed-type MCI following balloon DCP in the current study because we believe that it is easier to control and carries a lower risk than pulled-type MCI in conjunction with balloon DCP. In this study, we aimed to compare the surgical outcomes of antegrade balloon DCP with pushed-type MCI to balloon DCP alone in patients with complete PANDO.

## 2. Materials and Methods

We conducted a retrospective, non-randomized, comparative consecutive study in the Ophthalmology Department of National Cheng Kung University Hospital in Taiwan. Approval from the Institutional Review Board at National Cheng Kung University Hospital was obtained. All procedures in this study with human participants were in accordance with the ethical standards of National Cheng Kung University Hospital, the 1964 Helsinki declaration, its later amendments, and all laws in Taiwan. An informed consent was waived by the Institutional Review Board at National Cheng Kung University Hospital due to the retrospective nature of the study.

From December 2014 to May 2019, patients with complete nasolacrimal duct obstruction (NLDO) based on obstructed intra-sac irrigation and the symptom of epiphora were enrolled in the present study. Post-operative follow up was required to be more than 6 months. Thereafter, we excluded patients aged younger than 18 years old, and those with previous lacrimal procedure, prior ocular adnexal procedures, upper lacrimal drainage obstruction, punctal stenosis, or ocular infection. All patients underwent antegrade balloon DCP (LacriCATH^®^, QUEST Medical Inc., Allen, TX, USA, 3 mm balloon diameter, 15 mm in length) followed by a pushed monocanalicular silicone stent (Masterka^®^; FCI SAS, Paris, France) intubation for 1 month or balloon DCP alone as the primary treatment for complete PANDO. Balloon DCP was covered by National Health Insurance in Taiwan whereas pushed-type MCI was not and required an extra fee from patients for the MCI tube. Thus, surgical interventions whether to use the pushed-type MCI or not following balloon DCP in the current study was mainly determined by the patient’s decision. Subgroup analysis was performed based on age.

All procedures were performed by the same ophthalmologist, Dr. CCL, under local anesthesia at National Cheng-Kung University Hospital, Taiwan. The procedure of balloon DCP in the two groups was as follows. Initially, we dilated the upper punctum. Then, probing was performed with a Bowman probe. In the next step, we inserted a balloon catheter via the superior punctum until the superior mark of the catheter reached the lacrimal punctum (15 mm distance from the balloon). Next, we inflated the balloon using a manometer at eight atmospheres for 90 s followed by another one at the same pressure for 60 s. After the balloon was deflated, we withdrew the balloon catheter until the second mark of the catheter reached the lacrimal punctum (10 mm from the balloon). A 90-s inflation, deflation, and a 60-s inflation were performed again at eight atmospheres.

In the balloon DCP with MCI group, we then performed the tube-inserting procedures. After the balloon catheter was deflated and removed, we dilated the lower punctum and inserted a sizer through the inferior punctum until its tail reached the nasal floor, meanwhile testing whether the pushed-type antegrade MCI was appropriate for the patient. Next, we measured the proper length of the introducer (30, 35, 40 mm) according to the point on the sizer where the punctum was fitted. After measuring the proper length, we removed the sizer and performed a pushed-type antegrade MCI through the inferior punctum. Then, we withdrew the guide along the axis of the lacrimal sac. Finally, an anchoring plug was inserted into the vertical canaliculus. We placed the pushed-type MCI through the lower punctum under the assumption that more tear flow drains down the lower punctum by gravity. We thought that by placing the pushed-type MCI in the lower rather than upper punctum, the drainage would be improved more efficiently after the removal of the tube. Nevertheless, this assumption has not been confirmed so far.

All operated eyes were treated with topical 0.3% gentamicin and 0.1% betamethasone sodium phosphate four times per day for 2 weeks, followed by topical 4% sulfamethoxazole and 0.1% fluorometholone four times per day for 2 weeks. Postoperative follow up was arranged at approximately 1 week postoperatively. After that, we arranged an appointment at 1 month postoperatively to remove the stents at the outpatient department without anesthesia. Then, we performed intra-sac irrigation to check whether the lacrimal system was patent. Afterwards, we arranged a further outpatient department follow up at 3 months and 6 months postoperatively. Intra-sac irrigation was performed at the two follow-up appointments.

Surgical success was defined as a patent lacrimal draining system based on the intra-sac irrigation results and patient’s subjective improvement of symptoms. The surgical outcomes were recorded at 1 month, 3 months, and 6 months postoperatively. Patients who missed their postoperative appointments for intra-sac irrigation and arranged another outpatient department follow-up themselves were included in this study. If the intra-sac irrigation demonstrated patency over 6 months postoperatively, we defined it as surgical success at 3 months and 6 months postoperatively. If the patient was unable to tolerate epiphora and underwent another surgical intervention less than 6 months postoperatively, the outcome was considered a surgical failure at 6 months postoperatively.

A statistical analysis was completed using Mann–Whitney U tests and the Fisher’s exact test via software Statistical Product and Service Solutions (SPSS, Armonk, NY, USA), version 20.00. Differences were considered statically significant if the *p* value was less than 0.05.

## 3. Results

A total of 72 eyes of 56 patients with complete NLDO treated with balloon DCP with or without pushed type MCI were included in our study. A total of 37 eyes of 29 patients were treated with balloon DCP combined with pushed-type MCI, while 35 eyes of 28 patients were treated with balloon DCP alone. There were no intraoperative complications found in either group. The baseline characteristics of the two groups are provided in Table 1. No significant differences between age, gender, or laterality were found. There were also no between-group differences found in the proportion of patients younger than 65 years old and those at least 65 years old.

At 1 month postoperatively, the balloon DCP with MCI group had a significantly higher success rate than the balloon DCP alone group (89.2% vs. 62.9%, *p* = 0.009). However, no significant differences were found at 3 months (73.0% vs. 62.9%, *p* = 0.25) or 6 months (70.2% vs. 60.0%, *p* = 0.25) postoperatively despite the fact that the success rates of the balloon DCP with MCI group were higher than that of the balloon DCP alone group (Table 2). No postoperative complications were found in patients treated with balloon DCP and MCI; nevertheless, one eye in the balloon DCP alone group developed chronic dacryocystitis.

We conducted a subgroup analysis by dividing the patients into those under 65 years old and those aged at least 65 years old. The demographics and the results of the subgroup analysis are demonstrated in Table 3. No significant differences were found in age, gender, and laterality.

The success rates were significantly higher in balloon DCP with MCI group than in balloon DCP alone group in patients aged less than 65 years old at 1 month, 3 months, and 6 months postoperatively (81.8% vs. 9.1%, *p* = 0.001; 72.7% vs. 9.1%, *p* = 0.004; 72.7% vs. 9.1%, *p* = 0.004; Table 4). On the other hand, there were no significant between-group differences in success rate in those treated with balloon DCP with or without MCI in patients at least 65 years old at 1 month, 3 months, and 6 months postoperatively (92.3% vs. 87.5%, *p* = 0.46; 73.1% vs. 87.5%, *p* = 0.18; 69.2% vs. 83.3%, *p* = 0.20; Table 4).

## 4. Discussion

The nasolacrimal duct obstruction can be categorized into congenital or acquired according to the onset of age, and the management of the two are fundamentally different. The etiology of congenital nasolacrimal duct obstruction (CNLDO) is mostly the failure of nasolacrimal duct canalization, which is often in regard to mechanical obstruction at the valve of Hasner [20]. CNLDO affects approximately 20% of newborns worldwide [20]. Among them, around 96% of infants of CNLDO have spontaneous resolution of the symptoms in their first year of life [21]. Another study revealed a 60% spontaneous resolution after the children were 2 years old and a decreasing rate of spontaneous remission after they turned 18 months old [22]. Therefore, the current consensus of treating CNLDO is observation of the symptom of epiphora in the 1 year of age before further surgical intervention for CNLDO. According to our experience, antegrade balloon DCP followed by a short-term intubation with the pushed-type MCI is a high-potential surgical approach with a high success rate (96.77%) as a primary surgical treatment for CNLDO [19].

Acquired nasolacrimal duct obstruction is the blockage of the lacrimal system, resulting in epiphora, intermittent mucopurulent discharge, and mattering of the involved eye. It can be roughly classified as PANDO and SALDO [1]. SALDO is secondary to defined etiologies of obstructions, and is categorized into infectious, inflammatory, traumatic, mechanical, and neoplastic SALDO. The main treatment is to define and target the cause of the obstruction.

PANDO is defined as nasolacrimal duct obstruction of unknown etiology. Different from CNLDO and SALDO, the exact cause of PANDO is under investigation and seems to be multifactorial. PANDO may be a result of chronic inflammation, subsequent fibrosis, and further obstruction of the nasolacrimal duct. Some pathological studies have also shown fibrous formation secondary to chronic inflammation in the obstructed lacrimal drainage system [23]. The chronic inflammation and fibrosis may be the reason that the balloon DCP combined with pushed-type MCI in patients with PANDO had a lower success rate than that in patients with CNLDO. There is other reported pathophysiology of PANDO. Ectopic nasal epithelial cells in the nasolacrimal duct may also play a role in nasolacrimal duct obstruction [24] and the lacrimal duct-associated lymphoid tissue derangement may alter the immune response and could further lead to PANDO [25]. In the current study, we found PANDO mostly affected patients around their sixth and seventh decades, characterized with female predominance, which was compatible with previous studies [7,8]. It might result from the relatively smaller diameter of nasolacrimal duct in female patients [26,27], hormonal changes particularly in postmenopausal women [28], and derangements of lacrimal drainage-associated lymphoid tissue in chronic dacryocystitis [25].

Surgical interventions for PANDO can be divided into reconstructive surgery and bypass surgery [2]. DCR, a bypass surgery, is currently considered as the standard treatment for PANDO due to its high success rate, which was reported with approximately 90% [3,4,6,8]. When it comes to complete PANDO, DCR has also been suggested as the gold standard. A meta-analysis reported a success rate varied from 64 to 100% in external DCR and 84 to 94% in endoscopic DCR [29]. Over the past few decades, endoscopic DCR has grown in popularity. In comparison with external DCR, endoscopic DCR has the advantages of better cosmetic outcome and no facial scarring, a shorter wound recovery time and hospital stay, less blood loss during surgery, and better visualization of endonasal anatomy. Endoscopic DCR enables the operator to simultaneously correct nasal abnormalities like a deviated nasal septum and hypertrophic turbinate, and it is applicable for acute dacryocystitis [30,31,32,33,34,35]. Disadvantages of endoscopic DCR include more expensive equipment and steep learning curve that a thorough understanding of endonasal anatomy is required [30,32,33]. Our experience in endoscopic DCR with silicone stenting for complete PANDO showed a success rate of 82% in endoscopic DCR with pulled-type MCI and 87% in endoscopic DCR with pushed-type bicanalicular intubation (BCI) [8]. Furthermore, the subgroup analysis of our previous study showed no significant difference of the success rates in patients with or without previous dacryocystitis [8]. Although DCR possesses a better surgical outcome, it resolves the PANDO at the cost of invasiveness, such as osteotomy regardless of endonasal or external approach. In order to decrease the level of invasiveness and bleeding while operating; transcanalicular laser-assisted DCR was developed as an alternative to the conventional DCR by Michalos et al. [36]. However, Ayintap et al. revealed a lower success rate (62%) in transcanalicular laser-assisted DCR in complete PANDO patients younger than 65 years old at 2-year follow up [37].

To avoid osteotomy and decrease the invasiveness, reconstructive surgeries were attempted to manage PANDO. Kuchar and Steinkogler used antegrade balloon DCP combined with silicone stent intubation for 3 to 6 months for complete NLDO, and the success rate was 90% at 3 months and 70% at 6 months postoperatively [16]. In the present study, antegrade balloon DCP was combined with pushed-type MCI intubation for only 1 month, which is the shortest reported intubation duration to date [7,9,10,11,12,15,16,17,38]. We believed that a shorter intubation time would contribute to a lower rate of early stent loss and reduce possible complications related to stents, such as keratitis, canaliculitis, granulation formation, or intracanalicular stent migration. In our study, we placed the pushed-type MCI through the lower punctum into the nasolacrimal duct under the assumption that more tear flow drains down the lower punctum by gravity and, thus, drains more efficiently from the lower punctum. This was different from those described in previous studies by Fayet et al. [39], in which the authors favored inserting the intubation through the upper punctum to prevent the punctal plug of MCI from touching the cornea and causing keratitis. Nonetheless, there was no corneal irritation or keratitis reported after surgery in our study, indicating that the punctal plug of pushed-type MCI anchored on the lower punctum did not raise the risk of corneal damage by stents.

A randomized trial conducted by Andalib et al. compared bicanalicular intubation (BCI) and MCI for partial PANDO, revealing no significant differences in the surgical outcome between the two [15]. However, to date, there have been no comparative studies for complete PANDO. Mimura et al. reported a clinical success rate of 91.7% at 6 months postoperatively using BCI for complete PANDO without canaliculi involvement, while Inatani et al. reported a success rate of 68% [2,17]. However, surgical outcomes for MCI for complete PANDO have not yet been studied.

We favored MCI rather than BCI in the present study. Despite the use of prophylactic antibiotics, the postoperative infection rate with BCI for complete PANDO has been reported in previous studies to be as high as 9.5% [15,17]. Furthermore, a high punctal laceration rate (13.6%) was also reported by Kashkouli et al. [38]. Each case in our study presented with PANDO without canalicular involvements; therefore, bicanalicular stents may be unnecessary. Wladis et al. reported a risk of damage to the uninvolved canaliculus with the BCI [40]. As a result, pushed-type MCI seemed to be a better option in our study due to adequate intubation of the nasolacrimal duct, easy and speedy insertion, and less discomfort for the patients. However, unlike MCI, bicanalicular stents can be fixed without a punctal plug. Therefore, if the surgeons considered the opening of the punctum to be inadequate for the punctal plug to be properly anchored in MCI, the BCI may be the preferred choice.

There are two currently available types of MCI: a pulled-type and a pushed-type MCI. The pulled-type MCI required some interventions at the inferior turbinate while the pushed-type one did not [41]. We favored pushed-type MCI stents because repeated manipulations at the inferior meatus were not required during intubation when comparing with pulled-type MCI or pulled-type BCI. This may reduce operation time, lessen intra-operative nasal bleeding, and minimize pain for patients.

Our success rate at 3 months postoperatively seemed to be lower when compared with a previous study in which antegrade balloon DCP with silicone stent intubation was combined (73.0% vs. 90%, respectively), but the result was similar at 6 months postoperatively (70.2% vs. 70%) [16]. Since a longer postoperative follow-up period has been suggested, the differences between previous studies and the current study at 3 months and 6 months postoperatively were not significant [16]. Furthermore, the current study revealed that the additional pushed-type MCI for antegrade balloon DCP did not significantly improve the surgical outcomes in general since the success rates were not significantly higher at 3 months and 6 months postoperatively despite the significantly higher success rate at 1 month postoperatively.

To our knowledge, our study is the first to compare the differences in success rates between patients younger than 65 years old and those older than 65 for balloon DCP and silicone stent intubation. Interestingly, we found the success rates when combining balloon DCP with pushed-type MCI were always significantly higher than when using balloon DCP alone in patients younger than 65 years old regardless of the postoperative timing. On the other hand, there were no significant differences in the success rates between patients receiving balloon DCP with or without pushed-type MCI if the patients were aged over 65 years old. The silicone stent maintained the patency of the nasolacrimal duct [42], while balloon DCP only creates a perioperatively patent nasolacrimal duct (NLD). The stent intubation might weigh more in younger patients who possess stronger inflammatory reaction and more wound adhesion post-operatively than the older patients. By combining balloon DCP with pushed-type MCI, the patency of the NLD was maintained by avoiding excessive scarring, and consequently resulted in a better surgical outcome.

Due to the invasiveness of the bypass surgery, reconstructive surgery such as balloon DCP or stent intubation also plays a significant role in the management of PANDO. Although our overall long-term success rates with balloon DCP combined with pushed-type MCI (70.2%) did not achieve that using conventional DCR (around 90%), the procedure still yielded several advantages. First, no general anesthesia was required for the surgical method in the current study. Second, we minimized the degree of invasiveness by preserving the original nasolacrimal system instead of performing an osteotomy and largely decreased the operation time. Third, the present method did not require further manipulation of the inferior nasal meatus; thus, we decreased the possibility of significant bleeding. Last but not least, no obvious immediate postoperative complications were found after antegrade balloon DCP with the pushed-type MCI in the current study. Many patients, especially elderly individuals, may select balloon DCP with or without intubations instead of endoscopic DCR as the primary therapy for their nasolacrimal duct blockages due to the benefits indicated above.

In an attempt to improve the surgical success of treating NLDO, antimetabolites such as mitomycin-C have been used in adjunct with probing [43], external DCR [44,45], as well as endoscopic DCR [45], and have resulted in a better success rate with the application of mitomycin-C. However, there has been no consensus regarding the dosage of mitomycin-C to date. Tsai et al. reported that 89% of eyes treated with lacrimal probing with adjunctive low-dose mitomycin-C (0.2 mg/mL) irrigation achieved a patent nasolacrimal duct 9 months after lacrimal probing for epiphora in adults, and no complications were noted during the follow-up period [43]. Various concentrations and durations of mitomycin-C application in DCR were also reported and reviewed [45]. Because of the effect of inhibiting the synthesis of collagen by fibroblasts and subsequently reducing the fibrosis, applying adjunctive low-dose mitomycin-C may be a feasible and safe way to improve the surgical outcome of antegrade balloon DCP with or without stenting in our current study, but further study is needed for a more solid conclusion.

The current study was limited by its retrospective and non-randomized design. Additionally, we did not use any radiographic assistance to confirm the diagnosis of complete PANDO, for example, transcanalicular endoscopy or dacryocystography [13,16]. In addition, our case number was relatively small, which may limit the statistical power. Current results implied efficiency and few complications with antegrade balloon DCP followed by pushed-type MCI for patients with complete PANDO at 6 months postoperatively, and further large-scaled investigations with longer follow-up duration are needed for a more definite long-term result.

In general, the combination of balloon DCP and pushed-type MCI did not lead to a significantly higher success rate as compared with balloon DCP alone for complete PANDO. However, in patients under 65 years old with complete PANDO, the combination of balloon DCP and pushed-type MCI was associated with significantly higher surgical success rates as compared with balloon DCP alone. In addition, the former procedure had several advantages over DCR, including its non-invasiveness and the avoidance of general anesthesia as well as hospitalization for postoperative care. Balloon DCP followed by pushed-type MCI could be of value when it comes to treating complete PANDO in younger patients and it could also serve as an alternative for the elderly who are poor candidates for general anesthesia.

## Figures and Tables

**Table 1 jpm-13-00564-t001:** Demographic data in patients in balloon DCP combined with pushed-type MCI group and balloon DCP alone group.

Variable	Balloon DCP with MCI (37 Eyes of 29 Patients)	Balloon DCP Alone (35 Eyes of 27 Patients)	*p* Value
Gender (male/female)	2/27	6/21	0.14 ^a^
Number of eyes			>0.99 ^a^
18–64 years old	11	11	
≥65 years old	26	24	
Mean age (years old)	69.3 ± 12.2	66.1 ± 13.2	0.23 ^b^
Laterality			0.24 ^a^
Right	19	23	
Left	18	12	

^a^ Two-tailed Fisher’s exact test; ^b^ Two-tailed Mann—Whitney U test.

**Table 2 jpm-13-00564-t002:** Postoperative surgical success rate in patients in balloon DCP combined with pushed-type MCI group and balloon DCP alone group.

	Balloon DCP with MCI	Balloon DCP Alone	*p* Value
1 month	33/37 (89.2%)	22/35 (62.9%)	<0.01 ^a^*
3 months	27/37 (73.0%)	22/35 (62.9%)	0.25 ^a^
6 months	26/37 (70.2%)	21/35 (60.0%)	0.25 ^a^

^a^ One-tailed Fisher’s exact test * *p* < 0.05.

**Table 3 jpm-13-00564-t003:** Demographic data in patients in balloon DCP combined with pushed-type MCI group and balloon DCP alone group for patients younger than 65 years old and aged at least 65 years old.

	Age <65 Years Old		*p* Value	Age ≥65 Years Old		*p* Value
	Balloon DCP with MCI	Balloon DCP		Balloon DCP with MCI	Balloon DCP	
Gender (male/female)	0/10	2/7	0.21 ^a^	2/17	4/14	0.40 ^a^
Mean age (years)	53.5 ± 8.8	49.8 ± 8.8	0.25 ^b^	76.0 ± 5.3	73.6 ± 6.3	0.12 ^b^
Laterality			>0.99 ^a^			0.27 ^a^
Right	6	7		13	16	
Left	5	4		13	8	

^a^ Two-tailed Fisher’s exact test; ^b^ Two-tailed Mann–Whitney U test.

**Table 4 jpm-13-00564-t004:** Postoperative surgical success rate in patients in balloon DCP combined with pushed-type MCI group and balloon DCP alone group for patients younger than 65 years old and aged at least 65 years old.

	Age <65 Years Old		*p* Value	Age ≥65 Years Old		*p* Value
	Balloon DCP with MCI	Balloon DCP		Balloon DCP with MCI	Balloon DCP	
1 month	9/11 (81.8%)	1/11 (9.1%)	<0.01 ^a*^	24/26 (92.3%)	21/24 (87.5%)	0.46 ^a^
3 months	8/11 (72.7%)	1/11 (9.1%)	<0.01 ^a*^	19/26 (73.1%)	21/24 (87.5%)	0.18 ^a^
6 months	8/11 (72.7%)	1/11 (9.1%)	<0.01 ^a*^	18/26 (69.2%)	20/24 (83.3%)	0.20 ^a^

^a^ One-tailed Fisher’s exact test. * *p* < 0.05.

## Data Availability

The datasets in the current study will be available from the corresponding author on reasonable request.

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
