# Peer review of "Balloon Dacryocystoplasty with Pushed Monocanalicular Intubation as a Primary Management for Primary Acquired Nasolacrimal Duct Obstruction"

_jpm, 2023, doi:10.3390/jpm13030564_

Round 1
Reviewer 1 Report
The follow of 6 months is short for this type of study. It is a retrospective study, try to get more follow-up duration even on the cost of the number of patients.
The introduction should be more lucid and focused on the topic.
The methods should include specificities of the Balloon (dimensions (2-3 mm), rigid straight balloon ), etc.
Any comments on the simultaneous use of Mitomycin -C to improve the duration of patency?
Author Response
Dear Reviewer:
Thanks for your valuable comments and suggestions.
- Comment:
The follow of 6 months is short for this type of study. It is a retrospective study, try to get more follow-up duration even on the cost of the number of patients.
Response:
We understood the short follow-up duration is the limitation of our study, and we encountered some difficulties in data collection for a longer follow-up duration. Since NCKUH is a tertiary referral medical center in Taiwan, most patients lost follow-up or requested to be referred to a local medical doctor after surgery and the epiphora resolved. We had telephoned the patients during data collection in this study, and most patients replied with symptoms relief and satisfactory results, but the willingness of outpatient clinic follow-up at our hospital is low due to the cost of time and money of an appointment in a tertiary center. We agree longer follow-up duration with larger sample size is needed for a more convincing conclusion, and we will keep working on more cases to collect more data for a larger number of patients and longer follow-up duration in the future.
- Comment:
The introduction should be more lucid and focused on the topic.
Response:
The introduction is revised accordingly.
“Although dacryocystorhinostomy (DCR) has been the gold standard treatment for primary nasolacrimal duct obstruction (PANDO) due to its high success rate [1-6], there are some attempts to avoid osteotomy and perform less invasive procedures to manage nasolacrimal duct obstruction, including balloon dacryocystoplasty (DCP) and silicone stent intubation. Current studies on balloon DCP, silicone stent intubation or a combination of the two procedures are mostly focused on partial PANDO, where the reported success rates varied, ranging from 25% to 76% [7-13]. Some studies have discussed balloon DCP for complete PANDO [11,12,14], while limited studies have discussed about silicone stent intubation [14-16], and even fewer studies reported the result of combination procedure in complete PANDO.
In this study, we aimed to compare the surgical outcomes of antegrade balloon dacryocystoplasty (DCP) with pushed monocanalicular intubation (MCI) to balloon DCP alone in patients with complete PANDO.”
- Comment:
The methods should include specificities of the Balloon (dimensions (2-3 mm), rigid straight balloon ), etc.
Response:
The detail of the balloon was elucidated in Methods.
“…All patients underwent antegrade balloon DCP (LacriCATH®, QUEST Medical Inc., Texas, USA, 3mm balloon diameter, 15mm in length)…”
- Comment:
Any comments on the simultaneous use of Mitomycin -C to improve the duration of patency?
Response:
We do think simultaneous use of MMC might be a promising method in managing wound healing and scarring in balloon DCP or silicone stenting. Tsai et al. (2002), Tabatabaie et al. (2007), and Masoomian et al. (2021) presented the possibility of MMC use in NLDO surgeries. Initially, the potential toxicities (limbus cell insufficiency, corneal epithelial toxicity, etc.) and the lack of experience let us hold back from the new attempts, yet more evidence and practical, detailed procedure were demonstrated, and we are willing to try simultaneous 0.02% MMC lacrimal irrigation in such procedures in recent future.
Reviewer 2 Report
In this study the authors describe their experience in the management of the nasolacrimal obstruction with balloon dacryocystoplasty with and without intubation. Below are my remarks:
· Page 2, line 68: the complete name of the surgeon is not necessary. The initials will suffice
· Page 2, line 73: the mark is distal rather than proximal to the tip of the balloon
· Page 5, lines 188-195: the two sentences needs rephrasing
· Page 5. Lines 204-208: the sentence needs rephrasing
Overall it is a well presented study with useful and clear information on the topic in discussion
Author Response
Dear Reviewer:
Thank you for your time and valuable comments.
- Comment:
Page 2, line 68: the complete name of the surgeon is not necessary. The initials will suffice
Response:
The complete name of the surgeon was revised into initials. (“CCL”)
- Comment:
Page 2, line 73: the mark is distal rather than proximal to the tip of the balloon
Response:
The marking detail was revised.
“…In the next step, we inserted a balloon catheter via the superior punctum until the superior mark of the catheter reached the lacrimal punctum (15 mm distal to the beginning of the balloon)…”
- Comment:
Page 5, lines 188-195: the two sentences needs rephrasing
Response:
Page 5 line 188-195 was rephrased.
“…The silicone stent maintained the patency of the nasolacrimal duct [19], while balloon DCP only creates a perioperatively patent nasolacrimal duct (NLD). The stent intubation might weigh more in younger patients who possess stronger inflammatory reaction and more wound adhesion post-operatively than the older patients. By combining balloon DCP with pushed-type MCI, the patency of the NLD was maintained by avoiding excessive scarring, and consequently resulted in a better surgical outcome….”
- Comment:
Page 5. Lines 204-208: the sentence needs rephrasing
Response:
Page 5. Lines 204-208 were rephrased:
“…However, Ayintap et al. revealed a lower success rate (62%) in transcanalicular laser-assisted DCR in complete PANDO patients younger than 65 years old at 2-year follow-up than that in the present study (72.7% at 6-month follow-up) [21]….”
Reviewer 3 Report
there should be a discussion and comparison to endoscopic DCR
Author Response
Dear Reviewer:
Thanks for your valuable comments and suggestions. We appreciate your time and effort to improve this paper.
Comment:
There should be a discussion and comparison to endoscopic DCR
Response:
We have added a few discussion of DCR and provided our previous experience of endoscopic DCR in line 206-210:
“…A meta-analysis reported a success rate varied from 64 to 100% in external DCR and 84 to 94% in endoscopic DCR. [20] Our experience in endoscopic DCR with silicone stenting for complete PANDO showed a success rate of 82% in endoscopic DCR with pulled MCI and 87% in endoscopic DCR with pushed bicanalicular intubation (BCI).[6]…”
In paragraph 5 and 6 of Discussion (from line 204-230), we discussed the pros and cons of DCR and the reason of the clinical decision to balloon DCP rather than DCR.
We added some perspective of applying mitomycin-C in line 231-236 since another reviewer raised the idea and inquired about our opinion of a perioperative MMC in such cases.
(“…In the attempt to improve the surgical success of treating NLDO, mitomycin-C has been used in adjunct with probing [23], external DCR [24,25] as well as endoscopic DCR [25], and results in a better success rate with the application of mitomycin-C. Applying adjunctive mitomycin-C may be a feasible way to improve the surgical outcome of antegrade balloon DCP with or without stenting in current study, but further study is needed for a more solid conclusion…”)
Another minor revision in line 79 and 82 was from the academic editor. We revised the description of the marking of the balloon catheter accordingly to making it more comprehensive.
Thank you again for your constructive comment.
